# Genetic Variation Study of Several Romanian Pepper (*Capsicum annuum* L.) Varieties Revealed by Molecular Markers and Whole Genome Resequencing

**DOI:** 10.3390/ijms252211897

**Published:** 2024-11-05

**Authors:** Anca Amalia Udriște, Mihaela Iordăchescu, Liliana Bădulescu

**Affiliations:** 1Laboratory of Plant Molecular Physiology, Research Center for Studies of Food Quality and Agricultural Products, University of Agronomic Sciences and Veterinary Medicine (UASVM), 011464 Bucharest, Romania; amalia.udriste@qlab.usamv.ro; 2Laboratory of Plant Molecular Biology, Research Center for Studies of Food Quality and Agricultural Products, University of Agronomic Sciences and Veterinary Medicine (UASVM), 011464 Bucharest, Romania; mihaela.iordachescu@qlab.usamv.ro; 3Faculty of Horticulture, University of Agronomic Sciences and Veterinary Medicine (UASVM), 011464 Bucharest, Romania

**Keywords:** *Capsicum annuum*, south-east Romania, molecular markers, Sanger sequencing, whole genome resequencing

## Abstract

Numerous varieties of *Capsicum annuum* L. with multiple valuable traits, such as adaptation to biotic and abiotic stress factors, can be found in south-east Romania, well known for vegetable cultivation and an important area of biodiversity conservation. To obtain useful information about sustainable agriculture, management, and conservation of local pepper varieties, we analyzed the genetic diversity and conducted deep molecular characterization using whole genome resequencing (WGS) for variant/mutation detection. The pepper varieties used in the present study were registered by VRDS in the ISTIS catalog between 1974 and 2019 and maintained in conservative selection; however, no studies have been published yet using WGS analysis in order to characterize this specific germplasm. The genome sequences, annotation, and alignments provided in this study offer essential resources for genomic research as well as for future breeding efforts using the *C. annuum* local varieties.

## 1. Introduction

*Capsicum annuum*, commonly known as chillies, peppers, or bell peppers, is a plant species belonging to the genus *Capsicum* in the Solanaceae family, a diploid and self-pollinating crop that is native to southern North America and northern South America [1]. The chilli peppers fruits are not only used as spices and vegetables but also for medicinal purposes and are rich in vitamins A and C. Additionally, they are utilized as natural coloring agents in cosmetics and as active ingredients in host defense repellents [2]. Although *C. annuum* is widely cultivated in Romania due to its economic importance, most of the cultivars are of foreign origin. There are few local Romanian varieties, and most of them have limited yield potential [3]. Therefore, preserving and improving local pepper varieties requires evaluation of their degree of variation based on genetic characteristics and monitoring of the desired characteristics with valuable genotypes.

Molecular markers play a crucial role in plant genetics; they are essential for estimating the variability of plant varieties and species, helping to detect genetic relationships within plant genera [4,5]. In the case of *C. annuum*, inter-simple sequence repeat (ISSR) markers have been utilized to assess genetic diversity and population structure [6], to identify genetic homogeneity, and to generate molecular profiles to study genetic variability [7]. Several studies have used microsatellites (SSRs) to characterize and generate a molecular genetic map of SSR loci for *C. annuum* [8], to study genetic variability [9,10], or to design SSR primers that are transferable between *Capsicum* species [11]. Moreover, whole genome sequencing (WGS) data can provide extensive insights into the molecular mechanisms that link the association of genotypes with diseases and traits, identifying relevant variants and assessing their functionality [12,13,14].

The germplasm collection of Vegetable Research and Development Station (VRDS) Buzău, Romania, consists of 214 accessions of *C. annuum* L. grouped by degree of genetic stability, as follows: stable, advanced, and segregating [3]. Therefore, seven Romanian pepper (*C. annuum* L.) genotypes (‘Decebal’/gDEC, ‘Vladimir’/gVLA, ‘Galben Superior’/gGAL, ‘Splendens’/gSPL, ‘Cosmin’/gCOS, ‘Roial’/gROI, and ‘Cantemir’/gCAN) with valuable traits patented by the VRDS and registered in the Official Catalogue of Cultivated Plant Varieties in Romania (ISTIS) [15] were chosen for molecular characterization. Currently, no molecular data are available for these genotypes. The aim of this study was to generate data to be included in the database of Romanian cultivars’ genetic profiles. This resource aims to assist plant breeders in the future by providing a selection of genitors able to provide genes that encode desirable traits.

## 2. Results

### 2.1. Genetic Diversity Analysis

#### 2.1.1. SSR Analysis

Five pairs of genomic SSR molecular markers designed for *C. annuum* [16] were selected to identify the genetic profiles of seven *C. annuum* genotypes. Molecular variation by SSR markers generated a total of 14 alleles; total bands and polymorphism information content (PIC) values are presented in Table 1. Genetic diversity analysis and modified Roger’s genetic distances were calculated using BIO-R software (version R-3.3.1). The amplified alleles varied in size from 190 bp to 530 bp. All 14 fragments resulted in polymorphic profiles among the cultivars. The mean PIC value was 0.54; the highest was 0.72 (SSR-P5P6), and the lowest was for SSR-P9P10 (NA) (Table 1). Polymorphic and monomorphic alleles amplified with SSR markers are presented in Appendix A. The analysis of genetic profiles using SSR-PCR molecular markers led us to select the most common bands as specific markers for identifying homologous markers among these genotypes. To identify common genetic variations, the 220 bp SSR-PCR band achieved with SSR-P3P4 was selected for cloning, Sanger sequencing, and data analysis.

For each locus, the number of alleles and the number of rare alleles were calculated with BIO-R software. The highest values of rare alleles were detected for varieties gSPL (0.86) and gCOS (0.73) (Appendix A). Diversity analysis detected: % of polymorphic loci (0.31); expected heterozygosity (0.45); number of effective alleles (1.85); Shannon diversity index (0.93). The observed heterozygosity (Ho) for SSR markers ranged from 0.00 to 0.72, with a mean value of 0.54 (Figure 1). For SSR markers, it was calculated that the modified Roger’s genetic distances ranged from 0.44 to 1.00, with a short distance between hot peppers gDEC and gVLA (0.44) and more distance between the gDEC and gROI (0.77) genotypes. No distance was observed between the gDEC and gGAL genotypes, meaning that they are completely similar. Among the bell pepper, sweet long red pepper, and red fibster pepper varieties, similarity values registered a short distance between gCAN and gSPL (0.44). gDEC and gCAN, and gGAL and gCAN showed relatively long distances (0.89), meaning they are highly dissimilar. Varieties gSPL and gGAL, and gSPL and gDEC displayed a big genetic distance (1.00) (Appendix A). The dendrogram produced by hierarchical clustering offers insight into how similar or dissimilar the genotypes are based on the Modified Roger’s Distances. The SSR profile revealed two clusters with gVLA, gDEC, and gGAL very similar and another cluster with gROI, gSPL, gCAN, and gCOS with an agglomerative coefficient of 0.76 (Figure 1).

#### 2.1.2. ISSR Analysis

Eight selected ISSR primers [17] showed amplification product sizes ranging from 350 to 4000 bp (Appendix A); total bands and polymorphism information content (PIC) values are presented in Table 2. Using BIO-R software, genetic diversity analysis and Nei’s genetic distances were calculated. The supplementary data include the results from calculus per locus analysis: expected heterozygosity (He), number of effective alleles (Ae), and specificity of a marker in each allele (Spe Allele) (Appendix A). The highest values of rare alleles were detected for varieties gDEC (8.05) and gVLA (9.49) (Appendix A). To identify common genetic variations, we selected the distinct 1000 bp ISSR-PCR band from P26 for cloning, Sanger sequencing, and data analysis. 

Genetic diversity analysis calculated with BIO-R revealed the proportion of polymorphic loci (%P), and expected and observed heterozygosity are presented in Figure 2. Together, these ISSR primers amplified 401 bands in all seven tested samples. PIC values of polymorphic ISSR markers varied from 0.08 to 0.29 (mean: 0.23). The Nei’s genetic distance results provided a genetic distance for *C. annuum* ranging from 0.21 to 1.50, with a short distance between gCOS and gGAL (0.21) genotypes, indicating high genetic similarity and a longer distance between gCAN and gVLA (1.50) genotypes. Among the hot peppers, the range values show a distance of 0.43 between gDEC and gROI which is smaller than the distance of 0.63 between gDEC and gVLA genotypes. Among the bell pepper and red fibster pepper varieties, similarity values between genotypes ranged from 0.21 to 0.41, with gCOS and gGAL being close (0.21) and gCAN and gSPL more distant (0.41) (Appendix A). The ISSR profile revealed two clusters with an agglomerative coefficient of 0.45, where gDEC and gVLA were grouped in one cluster based on their similarity and the second cluster with gROI and gSPL in the first group and gGAL, gCOS, and gCAN in the second group (Figure 2).

Multidimensional scaling analysis (MDS) in 2D representing the distances among objects in a visual way were calculated with BIO-R software. MDS 2D variations for ISSR suggested a structure comprising related genotypes GAL, gCAN; gROI, gCOS, and gSPL, and unrelated genotypes gDEC and gVLA. MDS 2D variations for SSR suggested a structure corresponding to a combination as follows: gDEC and gGAL, gROI and gCAN, gSPL and gCOS at a higher distance, and unrelated genotype gVLA (Figure 3).

### 2.2. NGS Data Analysis and Quality Control

Whole genome sequencing (WGS) has become the most rapid and effective method of identifying genetic variations in individuals of the same species or between related species. The variation information such as single nucleotide polymorphism (SNP), insertion and deletion (InDel), copy number variation (CNV), and structural variation (SV) obtained through resequencing is used in population genetics research and genome-wide association studies (GWAS) to investigate the causes of diseases, to select plants for agricultural breeding programs, and to identify common genetic variations among populations (Novogene Co., Ltd., Cambridge, UK). Whole genome sequencing (WGS) of seven Romanian pepper varieties was performed via an Illumina platform (NGS) by Novogene Co., Ltd., Cambridge, UK. The sequencing data presented quality scores Q30 above 90% for all studied genotypes, which guarantee the accuracy and reliability of the sequencing data according to high standard procedures.

#### 2.2.1. Single Nucleotide Polymorphism (SNP) Detection and Annotation

The individual SNP variations were detected in all studied genotypes, yet their number and distribution within the genomes differentiate among the genotypes. SNPs located upstream, within 1 kb away from the transcription start site of the gene, showed a higher value on the gSPL genotype and a lower value on the gVLA genotype. Therefore, in the exonic region, the gSPL genotype presented the highest number of synonymous and non-synonymous SNP mutations (Table 3). The number of SNPs in different regions of the genome for genotype gSPL is presented in Figure 4. The lowest rate of non-synonymous SNPs mutations located in the exonic region (mutation with changing amino acid sequence) was observed for the genotypes gVLA and gROI, whereas the synonymous SNPs’ (without changing the amino acid sequence) lowest frequency was observed on the gCOS and gROI genotypes. For all genotypes studied, the number of non-synonymous SNPs was higher than the synonymous ones (Table 3). SNPs located within the intergenic region, transitions, and transversions showed the highest value on the gCOS genotype, as well as the total number of SNPs. The lowest number of total SNPs was observed in the gVLA genotype.

Stop gain mutations that lead to the introduction of stop codon at the variant site are about six times more frequent on all genotypes compared with stop loss mutations that lead to the removal of stop codon at the variant site. Moreover, the genotype gSPL presented the highest number of stop gain and stop loss exonic SNPs, and the gCAN genotype showed the lowest value. Approximately two out of three SNPs in all samples were transitions (ts), a point mutation that changes a purine nucleotide to another purine (A ↔ G) or a pyrimidine nucleotide to another pyrimidine (C ↔ T), compared with transversions (tv), which are the substitution of a (two ring) purine for a (one ring) pyrimidine or vice versa. Among all genotypes, the most common SNP mutation type distribution was C:G > T:A and T:A > C:G, with the highest values for gCOS and gSPL, while C:G > G:C, T:A > A:T, and T:A > G:C mutation types were at the lowest values for the gVLA and gROI genotypes (Figure 4). SNP density per chromosome for the genotype gSPL had the highest density on chromosomes 9 (NC_029985.1), 10 (NC_029986.1), and 11(NC_029987.1) (Figure 5).

#### 2.2.2. Insertion/Deletion (InDel) Detection and Annotation

InDel variations were observed across all the genotypes studied, and their number and annotation discriminated between genotypes. The InDels were distributed in all regions of the genomes: upstream, exonic (stop gain, stop loss, synonymous, non-synonymous), intronic, splicing, downstream, upstream/downstream, intergenic, and others. The number of upstream InDels located within 1 kb away from the transcription start site of the gene was higher in all samples compared with downstream of the gene region. The genotype gDEC presented the highest number of frameshift InDel mutations that confer changing the open reading frame with a deletion or insertion. The lowest number of non-frameshift InDel mutations without changing the open reading frame with deletion or insertion sequences of three or multiple of three bases was detected in the genotype gVLA. The highest number of InDels was observed between the 1–3 bp insertion/deletion and decreased sharply after the 6 bp sequence length (Figure 6). The highest number of insertions, deletions, and intergenic mutations located within the >2 kb intergenic region was observed in genotype gCOS; therefore, the total number of InDel mutations was allocated to the gCOS genotype, and the lowest number to the gVLA genotype. InDel density per chromosome for genotype gCOS is presented in Figure 7. Statistics of InDels detection and annotation based on WGS for all studied genotypes are presented in Appendix A.

#### 2.2.3. Structural Variant (SV) Detection and Annotation

Structural variants (SVs) are genomic variations with mutations of relatively larger size (>50 bp), including deletions, duplications, insertions, inversions, and translocations. In all analyzed genotypes, the number of SVs located in the exonic region was approximately five times higher than those located in the intronic region. Genotype gCOS showed the highest number of total SVs located within the >2 kb intergenic region, deletions, inversions, intra-chromosomal translocations, and inter-chromosomal translocations. Hence, gCOS and gGAL presented the highest number of insertions (INS). The lowest number of INS was observed in the gVLA and gROI genotypes (Figure 8). A visual representation of these data on the statistics of SV detection and annotation based on WGS for all studied genotypes is presented in Appendix A.

#### 2.2.4. Copy Number Variation (CNV) Detection and Annotation

Copy number variation (CNV) is a type of structural variation showing deletions or duplications in the genome. The highest number of CNVs with increased copy number (duplications) and CNVs located in the exonic region was observed in the gVLA and gSPL genotypes. The gDEC and gCAN genotypes showed the highest number of upstream/downstream CNVs located within the <2 kb intergenic region and also the highest number of CNVs with decreased copy number (deletions). The total number of CNVs was assigned to the gDEC genotype and the lowest number to the gVLA genotype (Figure 9). Statistics of CNV detection and annotation based on WGS for all studied genotypes are presented in Appendix A.

### 2.3. Sanger Sequencing and Multiple Genomic Alignments

Genetic profiles analyzed through ISSR and SSR-PCR fingerprints prompted us to use the most common bands as specific markers in order to identify homologous markers within these genotypes. Six out of seven DNA bands (gVLA, gGAL, gSPL, gCOS, gROI, gCAN) with the size of 1000 bp previously obtained by ISSR-PCR (P26 primer) and seven DNA bands (gDEC, gVLA, gGAL, gSPL, gCOS, gROI, gCAN) of 220 bp obtained by SSR-PCR (P3/P4 primers) were amplified by PCR. The 1000 bp and 220 bp DNA fragments were gel purified and then cloned into the pCR™4-TOPO™ TA vector. The cloning process was confirmed by colony PCR and then sent for Sanger sequencing.

A genomic BLAST search of the 1000 bp ISSR-PCR fragment against the pepper WGS *Capsicum annuum* L. (taxid. 4072) produced significant alignments with LTR retrotransposon (query cover 94% and percent identity 83.47%). In order to find the localization on chromosomes, we performed an alignment of the ISSR-PCR marker against the *Capsicum annuum* reference genome UCD-10X-F1, and the result with a query cover of 99% and identity 99% was allocated to chromosome 12, UCD10Xv1.1 whole genome shotgun sequence. With NCBI Genome Workbench software version 3.8.2., a multiple genomic alignment was performed between the *C. annuum* reference genome Pepper Zunla 1 Ref_v1.0, the ISSR-PCR 1000 bp cloned fragment (LTR) UCD10Xv1.1 whole genome shotgun sequence, and seven BAM files of *Capsicum* annuum local genotypes sequences (gDEC, gVLA, gGAL, gSPL, gCOS, gROI, gCAN) from chromosome 12.

Hence, the ISSR-PCR 1000 bp band was amplified only in six out of seven probes because, in the gDEC genotype, the band was absent on the agarose gel. Genotype gDEC was highly mutated on chromosome 12, with more than fourteen SNPs only in the 1000 bp cloned PCR fragment, two of them close to the annealing primer sites (Figure 10). The BLAST search of the cloned marker against the reference genome Pepper Zunla 1 Ref_v1.0, unplaced genomic scaffold revealed transversions (tv) SNP mutations on the gSPL, gROI, and gCAN genotypes, with substitution of a purine for a pyrimidine (G ↔ T) on base position 41.494. In all genotypes, within the ISSR-PCR 1000 bp cloned fragment, an SNP mutation with substitution of a pyrimidine for a purine for (T ↔ G) on base position 41.809 was observed (Figure 10, Appendix A).

Nucleotide sequence analysis and database study revealed that the Sanger sequenced 220 bp DNA fragment amplified by SSR marker P3/P4 displayed significant sequence homology to *Capsicum annuum* pathogenesis-related protein 10 (PR-10) mRNA, complete CDS (query cover 100% and percent identity 100%). The BLAST search of the 220 bp DNA fragment sequence against *Capsicum annuum* reference genome UCD-10X-F1 assigned it to chromosome 3, whole genome shotgun sequence, with query cover 100% and identity 100%. The multiple genomic alignment of the *C. annuum* reference genome, SSR-PCR 220 bp cloned fragment (PR-10), and chromosome 3 sequences for all seven genotypes revealed transitions (ts) SNP mutations only on the gCOS genotype. On graphical sequence view, inside the SSR-PCR cloned fragment for the gCOS genotype, a point mutation that changes a purine nucleotide to another purine (G↔ A) on base position 254,256,561 and another SNP with changes from a pyrimidine nucleotide to another pyrimidine (T ↔ C) on base position 254,256,599 was observed. Genomic alignment revealed no SNP mutations inside the SSR-PCR cloned fragment for other genotypes. In addition, a trinucleotide insertion, CTT type, was observed on the 254,256,423 position for the gCAN genotype and one nucleotide insertion, T type, on the 254,256,512 position for the gCOS genotype (Figure 11, Appendix A).

For proper visualization of the structural variations on the whole genome, sequencing data of the gCOS genotype is presented according to mutation types with Circos plots. The outer ring represents the chromosomes, and inside the chromosomes ring are drawn the density of the SNP/InDel type distribution as well as for the SV/CNV type, the location, and size (Novogene Co., Ltd., Cambridge, UK.). For gCOS, the 90–200 M region on chromosome 4 (NC_029980.1) showed large deletions and inversions as well as translocations that involved chromosomes 5 (NC_029981.1), 6 (NC_029982.1), and 7 (NC_029983.1) (Figure 12). In addition, the 0–50 M region on chromosome 8 (NC_029984.1) showed large deletions and inversions that involved chromosome 10 (NC_029986.1). A strong relation between genomic positions of 150–200 M from chromosome 10 (NC_029986.1) is illustrated with 0–50 M from chromosome 12 (NC_029988.1) (Figure 12). A visual representation of WGS data with Circos plots, SNPs, and InDels density per chromosome for all studied genotypes is presented in Appendix A.

## 3. Discussion

In Romania, several types of pepper varieties (bell pepper, red fibster pepper, sweet long red pepper, or hot pepper) are traditionally cultivated for fresh consumption and for preserves [18]. Romanian consumers prefer locally grown vegetables for their taste, shape, color, and size, and pepper is one of the main vegetable crops preferred by local consumers, so breeders are constantly developing new varieties [19,20]. Despite this, there is a lack of molecular data and conservation status of these pepper varieties. The varieties were chosen not only for their superior organoleptic properties, good yield, and storage potential but also for their resistance to biotic and abiotic factors [21]. Studies regarding fruit quality [22], seed germination [23], fruit storage [24], the impact of environmental conditions on crop growing [25,26], and the influence of organic fertilizers [27,28] were performed on several Romanian varieties of pepper, including gGAL, gCOS, and gSPL. Several authors reported morphological and physiological studies [29] on seedling growth [30], but no molecular studies have been published yet using WGS analysis in order to characterize VRDS pepper germplasm. 

The dendrograms that show the hierarchical trees generated by software (BIO-R) exposed that all varieties were clearly separated into two clusters. The ISSR profile revealed two clusters where gDEC and gVLA were grouped in one cluster based on their similarity, and the second cluster with gROI and gSPL in the first group and gGAL, gCOS, and gCAN in the second group, might suggest certain relationships and possible gene exchange among these varieties (Figure 2). The SSR profile revealed two clusters with gVLA, gDEC, and gGAL that were very similar and another cluster with gROI, gSPL, gCAN, and gCOS (Figure 1). 

Genomic alignment of ISSR-PCR 1000 bp cloned fragment against the pepper reference genome produced significant alignments with LTR retrotransposons. LTRs are mobile genetic elements characterized by their long terminal repeats essential for transposable element integrations and some of the most abundant components found within eukaryotic genomes [31]. Genome-wide analysis of ISSR-PCR 1000 bp LTR retrotransposon showed 12 K distance from the LOC 107854643 serine/threonine–protein kinase *ATG1t* gene and 13 K distance from the LOC 124889513 *auxin-responsive protein SAUR68-like* gene, both on *Capsicum annuum* chromosome 12 UCD 10Xv1.1. 

Multiple alignments were conducted in the SOL Genomics Database with an alignment analyzer tool to identify potentially active LTRs within the Solanaceae family. A BLAST search against Solanaceae popular datasets revealed 99% identity of 1000 bp complete query length with *C. annuum* Dempsey V1.0, *C. annuum* Maor V1.0, *C. annuum* UCD10X, and *C. annuum* Zunla genomes, and 94% identity with *C. chinense* genome scaffolds (release 0.5). In the tomato reference genome (SL4.0), the tomato wild species *Solanum pimpinellifolium* L. LA1670 genome, and the *Solanum pennellii* Correll BAC ends, only a short fragment of 100 bp had 86% identity with an LTR ranging from 600–700 bp query length. Compared with the *Solanum tuberosum* L. genome and *Solanum tuberosum* L. Bac sequences, the same 100 bp short fragment from our clone had 88% identity, and an extra fragment of 200 bp with 79% identity was located at the 5′end of the 1000 bp cloned LTR fragment. In *Solanum virginianum* L. V 4.1, *Nicotiana benthamiana* Domin V 2.6.1., and *Petunia axillaris* (Lam.) Britton, Sterns & Poggenb. V 1.6.2. genomes, only a 200 bp fragment located at the 5′end had 80% identity. On the basis of sequence alignments, it seems that among the Solanaceous species, transposable elements can move through genomes and may have experienced distinct degeneration events along with genome evolutionary history [32,33]. 

The pepper genome, as a result of its large genome size, may be the best model for the analysis of genome expansion through the evolution of constitutive heterochromatic regions [34] and particularly LTR retrotransposons, which are major factors that constitute the heterochromatin sequences [35,36,37].

The database study revealed that the selected clone corresponding to the 220 bp SSR-PCR marker is an expressed sequence tag (EST)-SSR molecular marker identified from the transcribed region of the pathogenesis-related protein 10 (*PR-10*) gene. This EST-SSR molecular marker was present in all analyzed genotypes, indicating conservation of the wide-ranging PR-10 up-regulated protein expressed during hypersensitive response upon infection by pathogens. PR-10 are multifunctional proteins present throughout various plant tissues, playing a significant role in growth, development, and stress responses and a crucial role in plant defense against pathogens [38,39,40,41].

WGS analysis revealed that SNPs located upstream away from the transcription start site of the gene and SNP mutations in the exonic region (CDS) were the most abundant in the gSPL genotype. Moreover, the genotype gSPL presented the highest number of stop gain (introduction of stop codon at the variant site) and stop loss (removal of stop codon at the variant site) exonic SNP mutations. The genotype sSPL (Splendens, *Capsicum annuum* L. spp. *annuum* convar. *grossum* (L.) Filov. var. *tetragonum* Miller), red fibster pepper, is a variety registered in the ISTIS catalog in 2008. The lowest number of total SNPs was observed in the gVLA genotype (Vladimir, *Capsicumannuum* L. ssp. *annuum* convar. *microcarpum* Filov), a red hot pepper variety registered in the ISTIS catalog in 2015 (VRDS Buzău) [15].

SNPs located within the intergenic region, transitions, and transversions showed the highest value on the gCOS genotype, as well as the total number of SNPs. gCOS (Cosmin, *Capsicumannuum* L. spp. *annuum* convar. *longum* (DC.) Terpó) is a sweet long red pepper variety registered in the ISTIS catalog in 1984 (VRDS Buzău). SNPs are associated with variations in phenotype and resistance to disease, as they may alter protein structure and function, enhance the binding affinity of transcription factors, modify alternative splicing, and regulate non-coding RNA [42,43,44]. Additionally, SNPs have been used to identify QTLs underlying various traits in plants [45,46].

The highest density of insertions, deletions, and intergenic mutations located within the >2 kb intergenic region was observed in genotype gCOS and the lowest number in the gVLA genotype. InDels can affect the synthesis of proteins and functional RNA molecules. InDel mutations have been a valuable complement to SNPs and simple sequence repeats (SSRs) [47]. InDel variations can be formed by unequal crossover, transposable elements, and sequence replication in regions of repetitive DNA [45,48,49].

Structural variants (SVs) are genomic variations with mutations of relatively larger size (>50 bp) that can have significant effects on gene expression and phenotype [50]. Genotype gCOS showed the highest number of SVs located within the >2 kb intergenic region, deletions, inversions, intra-chromosomal translocations, and inter-chromosomal translocations. Hence, the total number of SVs was assigned to the gCOS genotype and the lowest one to the gVLA genotype. 

Copy number variation (CNV) is a type of structural variation that contributes to phenotypic variance, including duplications and deletions. CNVs, including duplications and deletions, can influence gene expression by disrupting gene coding sequences, perturbing long-range gene regulation, or altering gene dosage [51]. CNVs with increased copy number (duplications) and CNVs located in the exonic region were observed in the gVLA genotype. The gDEC genotype showed the highest number of upstream/downstream CNVs located within the <2 kb intergenic region and also the highest number of CNVs with decreased copy number (deletions). CNVs can influence gene expression by disrupting gene coding sequences and perturbing long-range gene regulation or altering gene dosage [51]. Circos uses a circular ideogram to facilitate the display of relationships between pairs of positions by the use of ribbons, which encode the position, size, and orientation of related genomic elements [52]. Whole genome variation distributions show aligned regions between chromosomes connected with ribbons to illustrate the relationships between genomic positions. gCOS displayed the highest number of SVs located within the >2 kb intergenic region, deletions, inversions, intra-chromosomal translocations, and inter-chromosomal translocations. Additionally, the width of the ribbon corresponded to the alignment length at specific locations (Figure 12).

## 4. Materials and Methods

### 4.1. Plant Material

Pepper seeds from seven Romanian pepper (*Capsicum annuum* L.) varieties as ‘Decebal’-gDEC (yellow hot pepper), ‘Vladimir’-gVLA (red hot pepper), ‘Galben Superior’-gGAL (yellow bell pepper), ‘Splendens’-gSPL (red fibster pepper), ‘Cosmin‘-gCOS (sweet long red pepper), ‘Roial‘-gROI (red hot pepper), and ‘Cantemir‘-gCAN (red bell pepper) were received from Vegetable Research and Development Station Buzău Station (South-East Romania) (Figure 13) and cultivated under greenhouse conditions (18–25 °C) in the Research Center for Studies of Food Quality and Agricultural Products of the University of Agronomic Sciences and Veterinary Medicine of Bucharest, Romania. 

### 4.2. DNA Extraction

Extraction of genomic DNA was performed using 100 mg young leaves for each of the seven Romanian varieties. Genomic DNA was extracted using an automated extraction system (InnuPure C16, Analytik Jena GmbH, Jena, Germany) based on the principle of magnetic particle separation for fully automated DNA isolation and purification. InnuPREP Plant DNA I Kit-IPC16 (Analytik Jena GmbH, Jena, Germany) was used for genomic DNA extraction following manufacturer’s instructions. A preliminary manual processing step was the external lysis of the starting material. The plant sample was ground in the presence of liquid nitrogen to a fine powder and homogenized with SLS lysis solution (containing CTAB as detergent component), Proteinase K, and RNase A solution. After external lysis, the extraction proceeded with automated DNA extraction using the Ext_Lysis_200_C16_04 program. The DNA was quantified at 260 nm, and its purity was measured at a 260 nm/280 nm absorbance ratio. All measurements were conducted with a NanoDrop TM1000 spectrophotometer (Thermo Fisher Scientific, Waltham, MA, USA), and DNA quality was also estimated in 1.2% agarose gels.

### 4.3. ISSR Analysis

Seven anchored ISSR primers consisting of di and tri-repeat motifs were selected [17] and synthesized by ANTISEL/CeMIA SA (cellular and molecular immunological applications, GR.) for screening in this study. These anchored primers have an extended portion of bases at the 5′ or 3′ end of their sequence to increase the specificity of the amplicon, such as polymorphic content, and their capability of distinguishing between genotypes (Table 2). Total volume for the PCR reactions was 25 µL, containing Platinum™ II Hot-Start PCR Master Mix (2x with Platinum™ II Taq Hot-Start DNA Polymerase premixed in an optimized PCR buffer with dNTPs and 1.5 mM MgCl_2_ in final reaction concentration (Invitrogen™, Carlsbad, CA, USA), 0.8 µmol/µL primer, 5 µL Platinum™ GC Enhancer, and 50 ng genomic DNA. Amplification was performed with a Mastercycler^®^ Nexus system (Eppendorf™SE, Hamburg, Germany) as follows: one cycle at 94 °C for 2 min, 35 cycles at 94 °C for 15 s, 51 °C for 30 s, 68 °C for 1 min, and a final extension of 68 °C for 2 min. Amplification products were separated in 1.5% agarose gels (TopVision-Thermo Scientific™, Waltham, MA USA) using 200 bp Ladder (Carl Roth^®^, Karlsruhe, Germany) and 1 kb Plus DNA ladder (Invitrogen™, Carlsbad, CA, USA) as reference. Gels were stained with 1x SYBR™ Safe DNA Gel Stain (Invitrogen™, Carlsbad, CA, USA) in 1x TAE buffer following a conventional protocol for electrophoresis. Agarose gels were scanned with a molecular imager PharosFX™ Plus (BioRad, Hercules, CA, USA) system at 488 nm and provided with an external laser for high-resolution and precise spectral assignment.

### 4.4. SSR Analysis

Six pepper genomic SSRs were selected [16] for local varieties screening and analyzed individually, as they have different melting temperatures (Table 1). PCR amplification was performed using a 20 µL reaction mixture containing Platinum™ II Hot-Start PCR Master Mix (2x) that contains Platinum™ II Taq Hot-Start DNA Polymerase premixed in an optimized PCR buffer with dNTPs and 1.5 mM MgCl_2_ in final reaction concentration (Invitrogen™) and 0.6 µmol/µL of each primer and 40 ng genomic DNA. The amplification was performed as follows: one cycle at 94 °C for 2 min, 35 cycles at 94 °C for 15 s, 49 °C for 1 min, 68 °C for 1 min, and a final extension of 68 °C for 10 min. The PCR products were analyzed with PharosFX™ Plus (BioRad) system, and the resulting molecular data were used to generate a cluster analysis.

### 4.5. Molecular Markers Data Analysis

Each ISSR and SSR band was classified as having polymorphic band present (“1”) or absent (“0”) for each sample and was typed into a computer file as a binary-matrix-like one for each molecular marker and treated as an independent locus. Only consistent bands were used in the analysis. The resulting molecular data were then analyzed by BIO-R software (Biodiversity analysis with R-Version 3.0) and a set of R programs that perform genetic diversity analysis of molecular data and calculate calculus per locus, calculus per genotype, expected heterozygosity, diversity among and within groups, Shannon index, number of effective alleles, proportion of polymorphic loci, Nei’s and modified Rogers’s distances, cluster analysis, and multidimensional scaling 2D plot and 3D plot [53]. Furthermore, observed heterozygosity (Ho) and polymorphism information content (PIC) of markers were calculated. The genetic distances were calculated based on analysis of the marker using Nei’s distance for ISSR markers and modified Roger’s distance by Wright–Malecot coefficient for SSR markers analysis [54]. Proportion of polymorphic gene was defined if the frequency of one of its alleles was less than or equal to 0.95 or 0.99 (Pj = q ≤ 0.95 o Pj = q ≤ 0.99), where Pj is the polymorphic rate and q is the frequency allele. This measure provides the criteria to determine whether a gene has variation [53]. Cluster analysis was performed using Ward’s minimum variance method, where the distance between two clusters is the ANOVA sum of squares between the two clusters added up over all the variables. Ward’s method tends to join clusters with a small number of observations and is strongly based on producing clusters with approximately the same number of observations. [53]. Observed heterozygosity (Ho) for markers, which is obtained by the ratio between the number of heterozygous individuals and the total number of individuals in the population, was calculated with the formula H = 1 − ∑(i = 0)^kp_i^2, where k is the number of alleles, and pi is the frequency of the i^th^ allele [55,56]. Polymorphism information content (PIC) of markers corresponds to its ability to detect polymorphism among individuals in a population. For dominant markers (ISSR), the PIC value indicates the probability of finding that marker in two different states (present or absent) in two randomly selected individuals in a population. Its value ranges from 0 for monomorphic markers to 0.5 for markers present in 50% of individuals and absent in the remaining 50% [56]. In the case of co-dominant markers, the PIC value was calculated in the same way as heterozygosity: PIC = 1 − ∑(i = 0)^kp_i^2 [57], where ‘k’ is the number of alleles, and pi is the frequency of the allele. For dominant markers, the following equation was used: PIC = 2f (1 − f) [58], in which ƒ is the frequency of present bands in the developing gel, and 1 − ƒ represents the frequency of absent bands [59].

### 4.6. NGS, Data Processing, and Sequencing Analysis

Whole genome sequencing (WGS) was performed via an Illumina platform (NGS) by Novogene Co., Ltd., Cambridge, UK. For library construction, the genomic DNA was randomly sheared into short fragments; obtained fragments were end-repaired, A-tailed, and further ligated with Illumina adapter. The fragments with adapters were PCR amplified, size selected, and purified. The library was checked with Qubit, real-time PCR was used for quantification, and bioanalyzer was used for size distribution detection. Quantified libraries were sequenced on Illumina platforms according to effective library concentration and data amount required. Raw data were stored in FASTQ (.fq) format files [60], which contained sequencing reads and corresponding base quality. Sequencing quality distribution required a quality score Q30 above 80%, and data results showed that Q30 was over 90% for all studied genotypes. The effective sequencing data was aligned with the reference sequence through BWA [61] software version 0.7.8-r455, and the mapping rate and coverage were counted according to the alignment results (Novogene Co., Ltd., Cambridge, UK). Reference genome was downloaded from NCBI (Pepper_Zunla_1_Ref_v1.0), and the mapping rates of samples reflected the similarity between each sample and the reference genome. For the current 2,935,884,163 bp reference genome, the mapping rate of each sample ranged from 98.42% to 98.62%, the average depths were between 10.81xand 9.78x, and 1x coverages ranged from 96.86% to 97.68%. This result is in the qualified normal range and may serve in subsequent variation detection and related analyses (Novogene Co., Ltd., Cambridge, UK). 

The SNPs and InDels variations were detected with SAMTOOLS software version 1.3.1. with the parameter ‘mpileup-m 2-F 0.002-d 1000’ [62] and followed by annotation using ANNOVAR software version 2015 Dec.14 [63]. BreakDancer [64] software version 1.4.4. was used to detect insertion (INS), deletion (DEL), inversion (INV), intra-chromosomal translocation (ITX), and inter-chromosomal translocation (CTX) mutations based on the reference genome mapping results and detected insert size. Based on the reads depth of the reference genome, CNVnator [65] was used to detect CNVs of potential deletions and duplications with the parameter ‘-call 100’. The detected CNVs were further annotated by ANNOVAR (Novogene Co., Ltd., Cambridge, UK).

### 4.7. Cloning and Sanger Sequencing

TOPO™ TA Cloning™ Kits (Invitrogen, Carlsbad, CA, USA ) for sequencing were used for the insertion of amplified PCR products into a plasmid vector for sequencing analysis. ISSR and SSR-PCR products were analyzed by agarose gel electrophoresis, and then selected bands were gel extracted for cloning into the pCR™4-TOPO™ TA vector(Invitrogen, Carlsbad, CA, USA) with specially designed sequencing primer sites. OneShot™TOP10 Chemically Competent *E. coli* cells (Invitrogen, Carlsbad, CA, USA) were transformed individually with the recombinant pCR™4-TOPO™ TA plasmid vector that carried the PCR selected bands. Transformed *E. coli* cells were selected on LB agar plates containing ampicillin (100 μg/mL). Resulting colonies were randomly picked and cultured overnight in LB medium containing 100 μg/mL ampicillin. The presence of the inserted fragment within the vector was detected by colony PCR, and then the plasmids were isolated from positive colonies using a miniprep procedure (PureLink™-Quick Plasmid Miniprep Kit, (Invitrogen, Carlsbad, CA, USA). The plasmids with the inserted PCR fragments were then Sanger sequenced by MACROGEN Europe using the sequencing primer sites (M13F/R). The nucleotide sequences were compared with the database sequences using the NCBI-BLAST program of the National Center for Biotechnology Information (USA gov.).

## 5. Conclusions

Common genetic variation is a fundamental aspect of genetic diversity within populations and essential for the adaptability of populations to changing environments. Genetic variations provide the raw material for natural selection, allowing populations to evolve over time. The genome sequences, annotation, and alignments provided in this study offer essential resources for genetic and genomic research as well as for future breeding efforts within this important plant family.

By recognizing these differences, breeders can overcome the challenges associated with interspecific crosses and successfully enhance desirable agronomic traits. The genome-wide identification of evolutionarily conserved regions, particularly in non-coding genomic regions, will enhance the discovery and characterization of functional and regulatory elements. Additionally, this information will facilitate the identification of candidate genes associated with important agronomic traits.

## Figures and Tables

**Figure 1 ijms-25-11897-f001:**
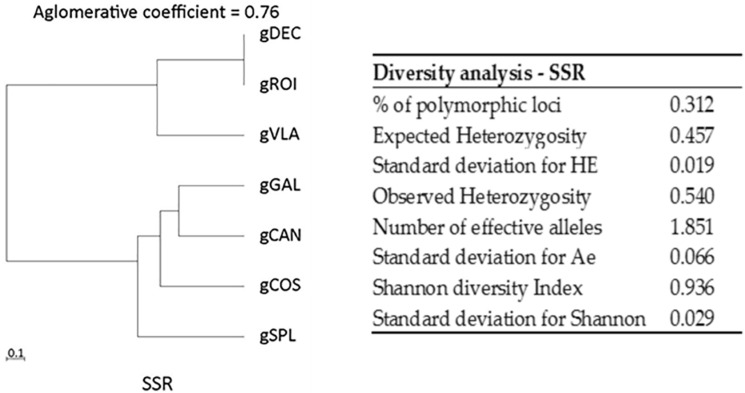
Dendrogram with agglomerative coefficients and diversity analysis for SSR markers.

**Figure 2 ijms-25-11897-f002:**
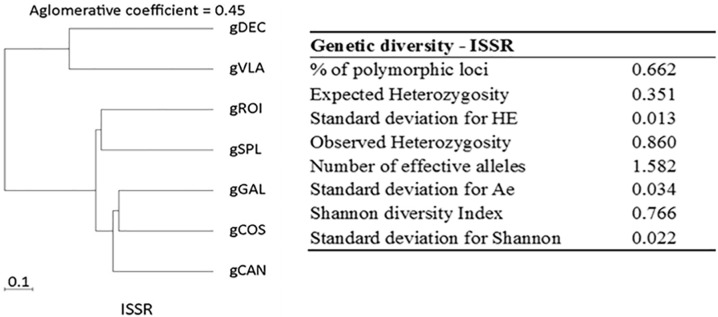
Dendrogram with agglomerative coefficients and diversity analysis for ISSR markers.

**Figure 3 ijms-25-11897-f003:**
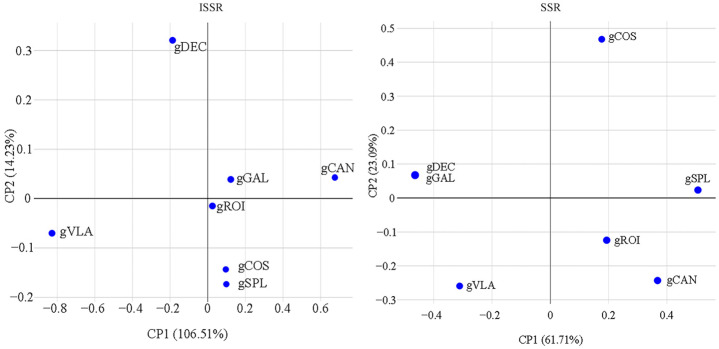
Multidimensional scaling analysis (MDS) for ISSR analysis based on Nei’s distance and SSR analysis based on modified Roger’s distance. CP1 and CP2 are the first and second principal coordinate matrices, respectively, in combination with a related genotype group: for ISSR analysis—gGAL, gCAN/gROI, gCOS, gSPL, and unrelated genotypes gDEC, gVLA are shown; for SSR analysis—gDEC, gGAL/gROI, gCAN/gSPL, gCOS, and unrelated genotype gVLA.

**Figure 4 ijms-25-11897-f004:**
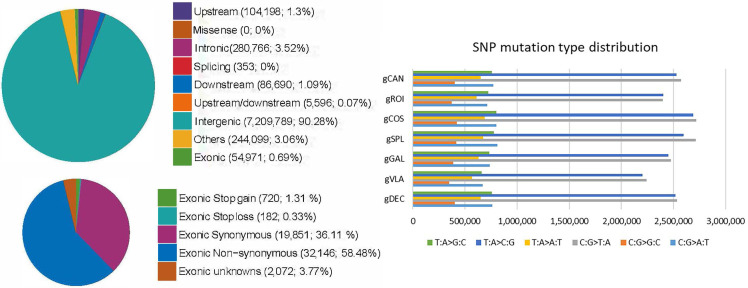
Frequency and type of SNP mutations for each genotype: the most common SNP mutation type distribution was C:G > T:A and T:A > C:G. The pie chart shows the number of SNPs and the SNP percentages in different regions of the genome for genotype gSPL; in the exonic region, the gSPL genotype presented the highest number of synonymous and non-synonymous SNP mutations.

**Figure 5 ijms-25-11897-f005:**
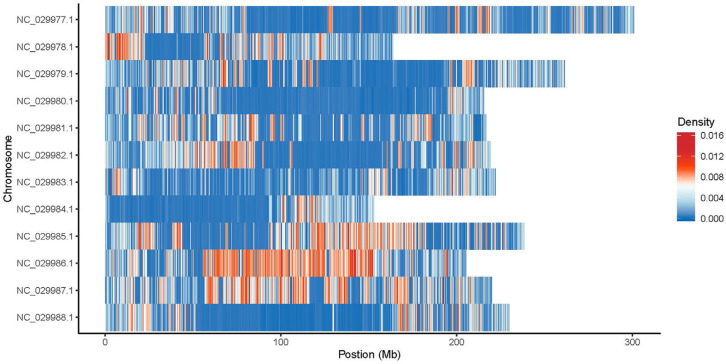
SNP density per chromosome for genotype gSPL.

**Figure 6 ijms-25-11897-f006:**
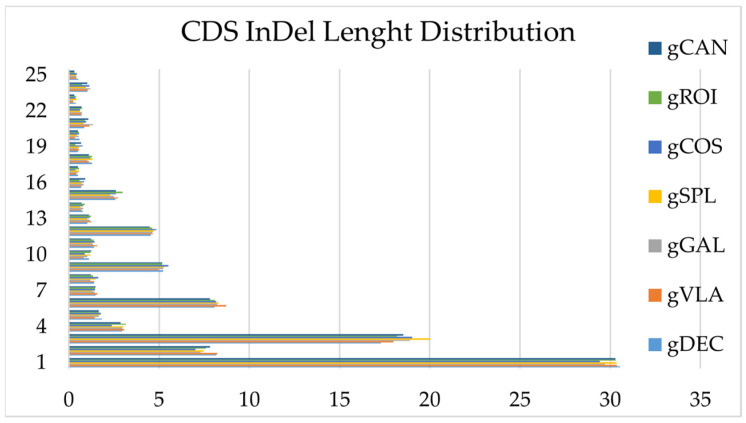
The length distribution of InDels for all genotypes within the coding sequence.

**Figure 7 ijms-25-11897-f007:**
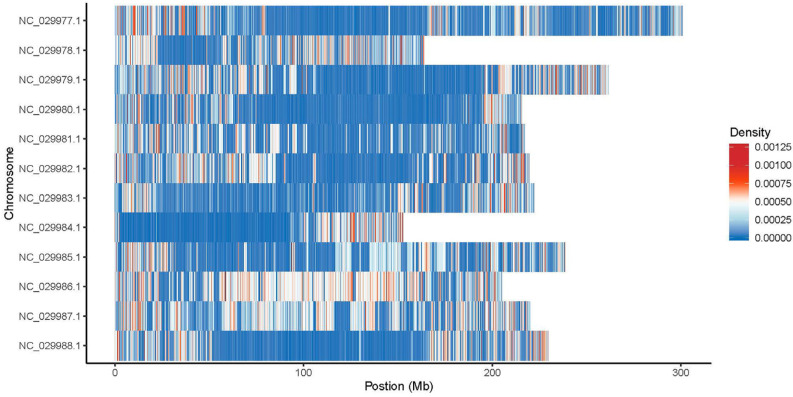
InDel density per chromosome for genotype gCOS.

**Figure 8 ijms-25-11897-f008:**
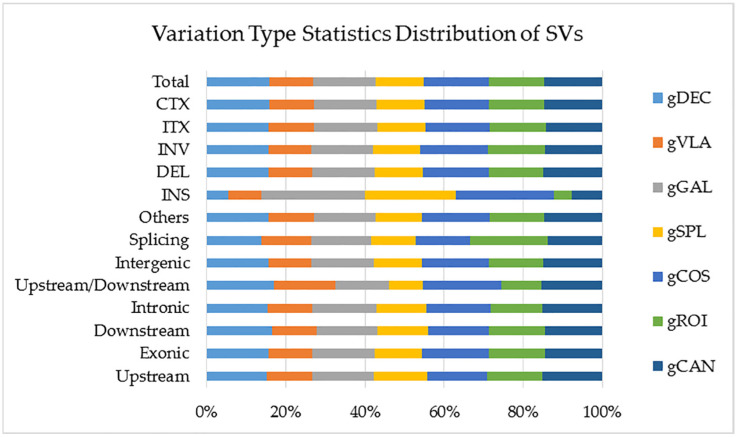
The number of SVs in different regions of the genome for all genotypes. gGAL and gCOS presented the highest number of insertions (INS). The lowest number of INS was observed in gVLA and gROI genotypes. The details of SV detection statistics are as follows: CTX (inter-chromosomal translocations); ITX (intra-chromosomal translocations); INS (insertion); DEL (deletion); INV (inversion); splicing; intergenic; upstream/downstream; intronic; downstream; exonic; upstream.

**Figure 9 ijms-25-11897-f009:**
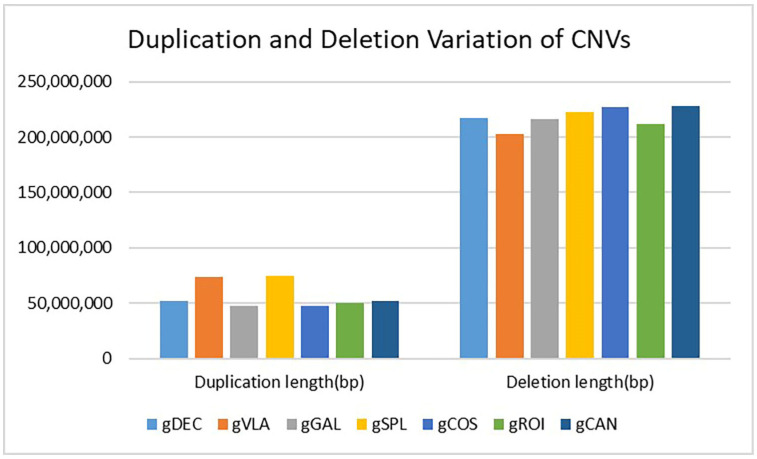
Variation type statistics distribution of CNVs in the genome.

**Figure 10 ijms-25-11897-f010:**
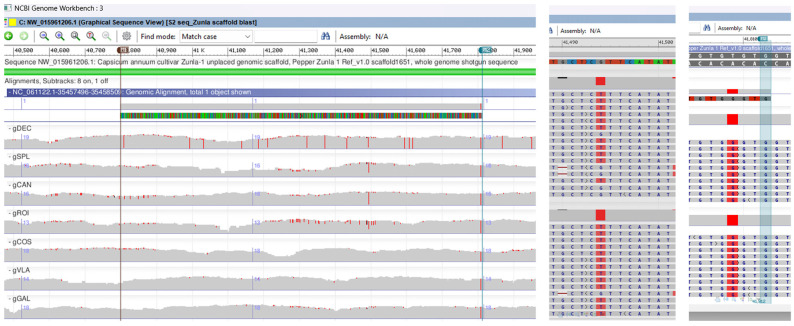
Multiple genomic alignment between the *C. annuum* reference genome Pepper Zunla 1 Ref_v1.0 unplaced genomic scaffold, the ISSR-PCR 1000 bp cloned fragment (LTR) UCD10Xv1.1 whole genome shotgun sequence ID: NC_061122.1, and all BAM files of *Capsicum annuum* local genotype sequences from chromosome 12. On the right side, the BLAST revealed SNP mutations on cloned fragments for gSPL, gCAN, and gROI genotypes on base position 41.494 and an SNP mutation on base position 41.809 for all seven genotypes. gDEC exhibits significant mutations on cloned fragments. On the right side is a close-up view of SNPs at specific positions.

**Figure 11 ijms-25-11897-f011:**
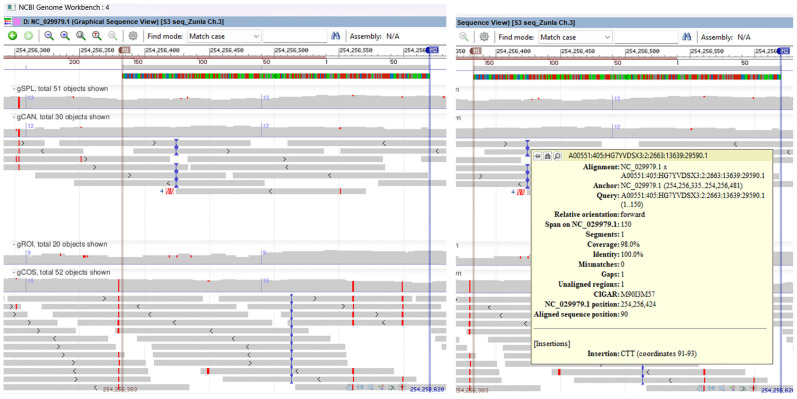
Multiple genomic alignment of *C. annuum* reference genome, SSR-PCR 220 bp cloned fragment (PR-10 protein), and chromosome 3 sequences for all seven genotypes revealed ts SNP mutations only on the gCOS genotype. On the left side is presented a graphical sequence view with a point mutation on base position 254,256,561 and another SNP on base position 254,256,599; on the right side, three nucleotide insertions as CTT type on the 254,256,423 position for the gCAN genotype and one nucleotide insertion as T type on the 254,256,512 position for the gCOS genotype.

**Figure 12 ijms-25-11897-f012:**
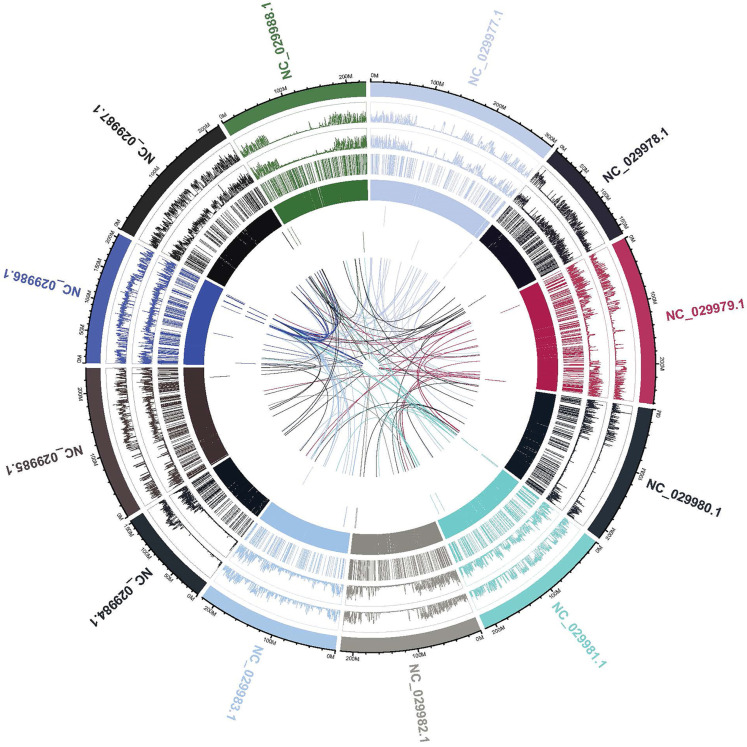
Visualization of the structural variations on the whole genome for gCOS genotype according to Circos plot analysis. The 90–200 Mb region on chromosome 4 (NC_029980.1) showed large deletions and inversions as well as translocations that involved chromosomes 5 (NC_029981.1), 6 (NC_029982.1), and 7 (NC_029983.1).

**Figure 13 ijms-25-11897-f013:**
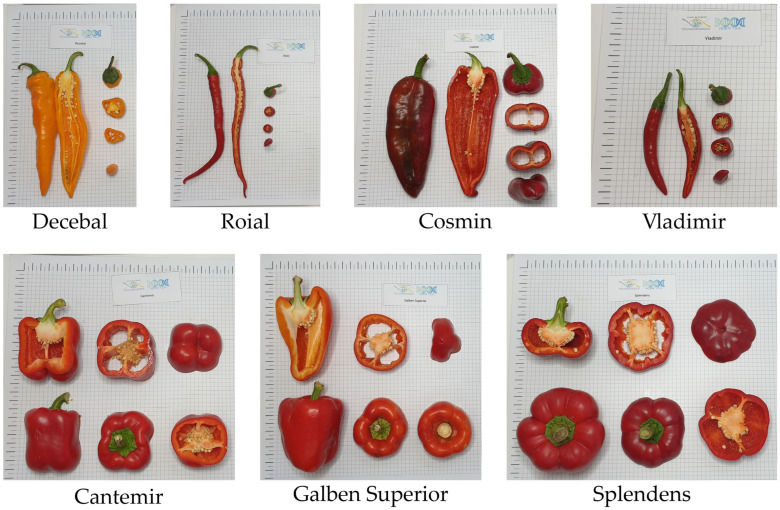
A visual representation of seven Romanian pepper (*C. annuum* L.) varieties with valuable traits, homologated populations by the VRDS, Decebal (gDEC), Vladimir (gVLA), Galben superior (gGAL), Splendens (gSPL), Cosmin (gCOS), Roial (gROI), and Cantemir (gCAN). Each label present in the individual pictures contain the logo of the Research Center for Studies of Food Quality and Agricultural Products (Centrul de Cercetare pentru Studiul Calității Produselor Agroalimentare), the logo of the research project (ADER 7.2.6.) and the variety’s name.

**Table 1 ijms-25-11897-t001:** Characteristics of SSR primers evaluated in *Capsicum* spp.

ID	Locus	Primer (Forward/Reverse)	Size (bp)	Tm (°C)	Total Alleles	PIC Value
SSRP3P4	AF244121	5′TACCTCCTCGCCAATCCTTCTG3′/ 5′TTGAAAGTTCTTTCCATGACAACC3′	200–400 bp	45	3	0.63
SSRP5P6	HpmS 1–148	5′GGCGGAGAAGAACTAGACGATTAGC3′/ 5′TCACCCAATCCACATAGACG3′	150–250 bp	45	4	0.72
SSRP9P10	HpmS 1_1	5′TCAACCCAATATTAAGGTCACTTCC3′/ 5′CCAGGCGGGGATTGTAGATG3′	260 pb	49	NA	NA
SSRP11P12	HpmS 1_274	5′TCCCAGACCCCTCGTGATAG3′/ 5′TCCTGCTCCTTCCACAACTG3′	190–530 bp	47	4	0.71
SSRP19P20	HpmS 1_172	5′GGGTTTGCATGATCTAAGCATTTT3′/ 5′CGCTGGAATGCATTGTCAAAGA3′	230–420 bp	48	3	0.66

**Table 2 ijms-25-11897-t002:** Characteristics of ISSR primers evaluated in *C. annuum* varieties.

ID	Primer	Tm (°C)	Total Bands (TB)	Range of the Amplification Product (bp)	PIC Value
P21	5′ACGACAGACAGACAGACA3′	51	38	850–4000 bp	0.08
P22	5′ACACACACACACACACCTG3′	50	28	500–2800 bp	NA
P23	5′GCAGACAGACAGACAGACGC3′	50	68	500–4000 bp	0.28
P24	5′GAGAGAGAGAGAGAGACTC 3′	50	56	800–3800 bp	0.23
P25	5′GAGAGAGAGAGAGAGACTC3′	50	80	550–3100 bp	0.29
P26	5′CACACACACACACACAAGT 3′	51	27	1000–2500 bp	0.26
P27	5′GACAGACAGACAGACAGT3′	51	70	380–4000 bp	0.20
P28	5′TCCTCCTCCTCCTCCAGCT3′	50	34	350–2700 bp	0.29

**Table 3 ijms-25-11897-t003:** Statistics of SNP detection and annotation based on WGS for all studied genotypes (Decebal/gDEC; Vladimir/gVLA; Galben Superior/gGAL; Splendens/gSPL; Cosmin/gCOS; Roial/gROI and Cantemir/gCAN). The details for SNP detection and annotation statistics are as follows: upstream (SNPs located within 1 kb upstream); exonic: SNPs located in exonic region (stop gain, stop loss, synonymous, non-synonymous); intronic; splicing; downstream; intergenic; transitions (ts); transversions (tv); heterozygous rate (Het. rate).

Genotype	gDEC	gVLA	gGAL	gSPL	gCOS	gROI	gCAN
Upstream	98,681	93,968	97,182	104,198	100,293	95,159	98,084
Exonic Stop gain	645	644	635	720	664	657	634
Exonic Stop loss	164	162	166	182	175	163	160
Exonic Synonymous	17,898	17,757	17,668	19,851	18,012	17,321	18,229
Exonic Non-synonymous	29,071	28,736	29,115	32,146	29,349	28,540	29,504
Intronic	255,395	240,657	249,296	280,766	262,290	246,344	260,011
Splicing	307	286	308	353	303	295	319
Downstream	80,621	77,439	79,503	86,690	82,171	77,640	80,567
Upstream/ Downstream	5010	4943	5126	5596	5365	4728	5058
Intergenic	6,899,284	6,009,182	6,706,315	7,209,789	7,383,834	6,528,591	6,955,410
Others	229,351	213,191	222,529	244,099	236,949	214,605	233,623
ts	5,051,787	4,442,276	4,923,303	5,313,137	5,408,262	4,796,597	5,099,410
tv	2,565,834	2,245,868	2,485,560	2,672,644	2,712,300	2,418,530	2,583,426
ts/tv	1.969	1.978	1.981	1.988	1.994	1.983	1.974
Het. rate	0.162	0.636	0.126	0.764	0.13	0.116	0.121
Total	7,617,621	6,688,144	7,408,863	7,985,781	8,120,562	7,215,127	7,682,836

## Data Availability

The original contributions presented in this study are included in the article/Appendix A. Further inquiries can be directed to the corresponding author(s).

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
