# Peer review of "Genetic Variation Study of Several Romanian Pepper (Capsicum annuum L.) Varieties Revealed by Molecular Markers and Whole Genome Resequencing"

_ijms, 2024, doi:10.3390/ijms252211897_

Round 1

Reviewer 1 Report

Comments and Suggestions for Authors

This manuscript presents results about the genetic diversity analysis of 7 genotypes of capsicum. 

It is not clear the aim of the work, but the manuscript seems to add different analyses without a clear aim. I do not understand why is necessary to perform SSR and ISSR and WGS? Why only 7 genotypes among the entire collection? The SSR analyses report 87 alleles in text for 5 markers among 7 samples, too much! In the table, the numbers are different.

ISSR and SSR are solved with agarose gel, that is not acceptable for the low resolution power.

WGS analysis is not open data. I did not find any reference about the deposit in an open source. There is no mention about genes studies, loci under selection, just the study about the fragments cloned obtained by ISSR and SSR markers.

The phenotypic data are totally absent, the number of genotypes analyzed is too low, and is difficult to understand the aim of this work. You have a large collection to study, would be more complete and interesting to analyze the genetic variability of it, starting with SSR and ISSR to reduce the number and choose the representative genetic pool to analyze with NGS.

Comments on the Quality of English Language

The english language is fine

Author Response

Comments 1: It is not clear the aim of the work, but the manuscript seems to add different analyses without a clear aim.

Response 1: Thank you for pointing this out. We agree with this comment and we have revised the introduction. The aim of the manuscript is to evaluate the genetic variability of local Capsicum genotypes and generate data that can be included in the database of Romanian cultivars’ genetic profiles. Currently, no molecular data is available for these genotypes. This resource aims to assist plant breeders in the future by providing a selection of genitors able to provide genes that encode desirable traits.

Comments 2: I do not understand why is necessary to perform SSR and ISSR and WGS?

Response 2: Thank you for bringing this into our attention. While it's true that it is not strictly necessary, we initiated SSR and ISSR analyses as a fast and cost-effective approach for assessing genetic variation. Then, we progressed to more in-depth molecular analyses with WGS for SNPs analysis in order to obtain comprehensive genetic information. These techniques are often used for different but complementary purposes in research, as breeding programs.

Comments 3: Why only 7 genotypes among the entire collection?

Response 3: Based on germplasm collection of Vegetable Research and Development Station (VRDS) Buzău, Romania, we have selected a limited number of varieties with superior organoleptic properties, storage potential, but also for their resistance to biotic and abiotic factors. The variation among the selected genotypes is as follows: yellow hot pepper (gDEC), red hot pepper (gVLA), yellow bell pepper (gGAL), red fibster pepper (gSPL), sweet long red pepper (gCOS), red hot pepper (gROI) and red bell pepper (gCAN). This study is part of a bigger project founded by the Romanian Ministry of Agriculture and Rural Development, and one of the goals of this project is to create a database of Romanian cultivars/varieties based on their economic potential.

Comments 4: The SSR analyses report 87 alleles in text for 5 markers among 7 samples, too much! In the table, the numbers are different.

Response 4: Thank you for pointing this out. We agree with this comment. Therefore, we have corrected the mistake, the correct number is that from the table (14 alleles).

Comments 5: ISSR and SSR are solved with agarose gel that is not acceptable for the low resolution power.

Response 5: Thank you for pointing this out. We choose to use agarose gels for visualizing amplicons because, in our particular case, we have observed differences between varieties. Moreover, we did gel extraction on specific bands and the Sanger sequencing revealed the correct size of our agarose gel fragments.

Comments 6: WGS analysis is not open data. I did not find any reference about the deposit in an open source. There is no mention about genes studies, loci under selection, just the study about the fragments cloned obtained by ISSR and SSR markers.

Response 6: Thank you for pointing this out. As we mentioned in acknowledgments, the Romanian Ministry of Agriculture and Rural Development funded the ADER 7.2.6 project, and for that we need its approval to upload any genomic data in an open source. We will prepare the documents and request the permission to upload some data.

Comments 7: The phenotypic data are totally absent, the number of genotypes analyzed is too low, and is difficult to understand the aim of this work.

Response 7: Thank you for this observation. We added in the Material and Methods section, at Plant material description a new Figure (Figure 13) with visual representation of the pepper varieties with valuable traits and a Supplementary Table (Table S12) with phenotypic data about genitors, patent numbers, seedlings monitoring, plant and fruit characteristics.

Reviewer 2 Report

Comments and Suggestions for Authors

The rewieved article "Genetic variation study of several Romanian pepper (Capsicum annuum L.) varieties revealed by molecular markers and whole genome resequencing" is interesting and in my opinion requires minor corrections.

The paper is well written and the only chapter, which needs improvement is Discusssion. In my opinion the repeated information from Introduction and Results especially in the first part of Discussion should be avoided. The authors also included Figures to the Discussion, which should be added to the Results. The whole botanical names of plants should be added (e.g. Petunia axillaris (Lam.) Britton, Stern & Poggenb.).

Author Response

Comments 1: In my opinion the repeated information from Introduction and Results especially in the first part of Discussion should be avoided.

  Response 1: Thank you for pointing this out. We agree with this comment and we have revised the Discussion.

Comments 2: The authors also included Figures to the Discussion, which should be added to the Results.

Response 2: Thank you for bringing this into our attention. Two figures from Discussion were moved to Results.

Comments 3: The whole botanical names of plants should be added (e.g. Petunia axillaris (Lam.) Britton, Stern & Poggenb.).

Response 3: Thank you for pointing this out. We agree with this comment. Therefore, we have corrected the botanical names.